# Definition of the Acceptor Substrate Binding Specificity in Plant Xyloglucan Endotransglycosylases Using Computational Chemistry

**DOI:** 10.3390/ijms231911838

**Published:** 2022-10-05

**Authors:** Barbora Stratilová, Eva Stratilová, Maria Hrmova, Stanislav Kozmon

**Affiliations:** 1Institute of Chemistry, Slovak Academy of Sciences, SK-84538 Bratislava, Slovakia; 2Jiangsu Collaborative Innovation Centre for Regional Modern Agriculture and Environmental Protection, School of Life Science, Huaiyin Normal University, Huai’an 223300, China; 3School of Agriculture, Food and Wine & Waite Research Institute, University of Adelaide, Glen Osmond, SA 5064, Australia; 4Medical Vision o.z., SK-82108 Bratislava, Slovakia

**Keywords:** binding free energy calculations, glycoside hydrolase family 16, homo- and hetero-transglycosylation reactions, molecular docking, molecular dynamics simulations, TmXET6.3, PttXET16A

## Abstract

Xyloglucan endotransglycosylases (XETs) play key roles in the remodelling and reconstruction of plant cell walls. These enzymes catalyse homo-transglycosylation reactions with xyloglucan-derived donor and acceptor substrates and hetero-transglycosylation reactions with a variety of structurally diverse polysaccharides. In this work, we describe the basis of acceptor substrate binding specificity in non-specific *Tropaeolum majus* (TmXET6.3) and specific *Populus tremula* x *tremuloides* (PttXET16A) XETs, using molecular docking and molecular dynamics (MD) simulations combined with binding free energy calculations. The data indicate that the enzyme-donor (xyloglucan heptaoligosaccharide or XG-OS7)/acceptor complexes with the linear acceptors, where a backbone consisted of glucose (Glc) moieties linked via (1,4)- or (1,3)-β-glycosidic linkages, were bound stably in the active sites of TmXET6.3 and PttXET16A. Conversely, the acceptors with the (1,6)-β-linked Glc moieties were bound stably in TmXET6.3 but not in PttXET16A. When in the (1,4)-β-linked Glc containing acceptors, the saccharide moieties were replaced with mannose or xylose, they bound stably in TmXET6.3 but lacked stability in PttXET16A. MD simulations of the XET-donor/acceptor complexes with acceptors derived from (1,4;1,3)-β-glucans highlighted the importance of (1,3)-β-glycosidic linkages and side chain positions in the acceptor substrates. Our findings explain the differences in acceptor binding specificity between non-specific and specific XETs and associate theoretical to experimental data.

## 1. Introduction

Xyloglucan endotransglycosylases (XETs) are fundamental glycosidic bond-forming biocatalysts that operate during the biogenesis of plant cell walls (CWs) and fulfill the structural and mechanistic roles in CW formation and remodelling. The action of XETs is irreplaceable in physiological cellular processes that underlie CW expansion and reconstruction [1,2,3,4,5]. The discovery of XETs from various plant sources was reported independently by three research groups [6,7,8], and since then, significant knowledge of the structure, function, biochemistry, biophysics, and evolutionary relationships of XETs has been acquired [9,10,11].

According to the Enzyme Commission (EC), XETs are classified among glycosyl transferases, with their systematic name xyloglucan:xyloglucosyl transferases (EC 2.4.1.207) [12]. This EC description recognises xyloglucan (XG) as a donor substrate and XG or XG-derived oligosaccharides (XG-OS) as acceptor substrates, which are utilised in homo-transglycosylation reactions. Concerning the focus of this work, it is important to note that the definition of XETs by EC still includes a comment ‘does not use cello-oligosaccharides as either donor or acceptor’. As it will be shown in this work, and in light of the current knowledge, this remark is obsolete [13,14,15,16,17,18,19].

The Carbohydrate-Active enZYmes Database (CAZy) [20] and CAZypedia [21] clarify the issue of the XET nomenclature and classify entries based on their tertiary structures, substrate specificity, phylogenomic relationships, and evolutionary history. According to CAZy, XETs are members of the glycoside hydrolase family 16 (GH16), while the transferase groups specifically contain the enzymes utilising ‘activated’ sugar phosphates as the glycosyl donors. The GH16 family is subdivided into 23 subfamilies according to the features in their tertiary structures [22]. XET enzymes are allocated together with xyloglucan endohydrolases (XEHs, EC 3.2.1.151) into the GH16_20 subfamily [23]. The structure-based studies of these enzymes [24] showed their close similarity in tertiary structures and provided evidence for the evolution of XEHs from XETs [23,25,26]. These findings were supported by a cross-genome survey of the evolutionary origin of endoglucanases within the GH16 family and XET/XEH enzymes [27]. The latest phylogenomic and comparative structural analyses [10] derived the origin of the GH16_20 subfamily from the non-plant alphaproteobacteria ExoK biocatalysts involved in the loosening of biofilms in icy environments [28], rather than from the previously suggested bacterial lichenases [29,30], which are classified in the GH16_21 subfamily [22]. Predicted intermediates between ExoKs and XETs are the charophycean EG16-2 enzymes, which originated due to the horizontal gene transfer event during the Cryogenian geological period [31,32,33].

In addition to homo-transglycosylation reactions catalysed by XETs, a new subtype of transglycosylation reactions was identified in 2006, when XETs from a crude nasturtium (*Tropaeolum majus*) extract were found to recognise poly- and oligosaccharide substrates other than XG-derived [13]. These reactions were in 2007 demonstrated in near-homogenous barley HvXET5 [14], and later in other plant enzymes [11,34] and crude extracts [9,16,17]. Currently, there are only a few XETs with defined primary structures for which enzyme activities on substrates other than XG were described [34]. In principle, the reactions catalysed by XETs are subdivided into three types, as described below.

*(i) Homo-transglycosylation reactions with XG-derived substrates*: The most appropriate representative of strictly specific XETs is the poplar PttXET16A [25,35], which exclusively recognises XG-derived substrates and is the best characterised XET due to its tertiary structure derived from X-ray crystallography. An enzyme with analogous substrate specificity, the *Pinus radiata* PrXTH1 [36], showed in-silico a weak interaction with cello-oligosaccharides (Cello-OS) as substrate donors [37]. Nevertheless, both PttXET16A and PrXTH1, according to phylogenomic analyses of the GH16 family [34], clustered with the HvXET5 enzyme. 

*(ii) Hetero-transglycosylation reactions with cellulose-derived or (1,4;1,3)-β-d-glucan (mixed-linkage glucan; MLG)-derived substrates*: As determined experimentally, HvXET5 in a near-homogenous form in-vitro catalysed the transfer of hydroxyethyl cellulose (HEC) fragments on XG-derived oligosaccharides (XG-OS; 44% efficiency) or XG fragments on Cello-OS. Here, the reaction rates were comparable to those of XG with XG-OS, while those with the MLG donor substrate were low (efficacy 0.2%) [14]. The formation of hybrid products was confirmed by mass spectrometry. 

The next near-homogenous XET, AtXTH3 from *Arabidopsis thaliana* L. Heynh, recognised cellulose as the donor substrate [38] and cellulose, Cello-OS, and XG-OS as acceptors, in addition to XG-derived substrates. Moreover, this enzyme formed cello-oligomers from the aminopyridyl derivative of cellohexaose with higher degrees of polymerisation (DPs) than the original substrate, as confirmed by mass spectrometry. In the absence of other substrates, insoluble cellulose-like material was formed. Notably, barley HvXET5 [14] and AtXTH3 [38] clustered within the same phylogenetic XTH I clade, as presumably XG-specific PttXET16A and PrXTH1 [25,36], although they segregated to different sub-clades [34]. 

Further XETs with a defined primary structure capable of transferring besides XG, also cellulose, or MLG fragments were described in the *Equisetum fluviatile* L. and were named EfXTH-A, EfXTH-H, and EfXTH-I [39]. The homo-transglycosylation activity (with the XG/XG-OS pair) was a dominant reaction for all acidic isoforms, whereas the efficiencies of transfers with cellulose and MLG fragments differed. Similarly, as HvXET5 [14], EfXTH-A showed a comparable transfer of MLG fragments to XG-OS (efficiency 0.2–0.3%), while the transfer with cellulose was incomparably higher with the barley HvXET5 isoform [14]. The hetero-transglycosylation activities of both EfXTH-H and EfXTH-I were equivalent to those of the MLG substrates, and the activity with the cellulose donor was around one order of magnitude higher than that of EfXTH-A [39]. On the contrary, other transglycosylating enzymes from *Equisetum*, the hetero-transglucanase (HTG) and MLG:xyloglucan endo-transglucosylase, preferred cellulose and MLG substrates with XG-OS as respective donors and acceptor substrates [40,41]. The predicted function of the latter enzyme was to reconstruct hemicelluloses in horsetail shoots [40]. Similarly, as EfXTHs, HTG was also subjected to molecular modelling [41]. Regardless of its donor specificity, HTG as a member of the GH16_20 subfamily, clustered within the XTH II clade [10,34]. It was suggested that XETs from the XTH II clade evolved from the XTH I clade catalysts [10]. The representatives of this clade first appeared in lycophytes, but HTG- and MLG:xyloglucan endotransglycosylase-like activities were also found in charophytic algae [9,42,43]. The atomic structures of these XET enzymes have yet to be determined. Among others, HEC and Cello-OS substrates served as respective donors and acceptor substrates, also for partially purified XETs from parsley roots [44] or for XETs isolated from parsley stems and leaves, and nasturtium stems, leaves, and roots [17]. The efficiency of transglycosylation did not exceed 5% with the HEC/Cello-OS pair compared to the XG/XG-OS pair.

*(iii) Hetero-transglycosylation reactions with acceptors other than XG-, cellulose-, or MLG-derived:* Unlike XG substrate donors, HEC or carboxymethyl cellulose derivatives, and MLG substrates, i.e., donors derived from polysaccharides with a backbone made of Glc moieties connected mainly by (1,4)-β-glycosidic linkages, the structure of substrate acceptors differed significantly both in terms of the saccharide moieties and glycosidic linkages that interconnect them. The broad acceptor specificity of XETs isolated from nasturtium germinating seed extracts [13] initiated the structural studies linked to the substrate specificity of the major *Tropaeolum majus* TmXET6.3 isoform (named according to its isoelectric point of 6.3), that clustered in the XTH II clade [17,34]. Recombinant and near-homogenous TmXET6.3 did not utilise polysaccharides other than XG or HEC as donors [17], but it was able to transfer their fragments to a whole spectrum of structurally different neutral acceptor substrates derived from cellulose (Cello-OS), MLG (MLG-OS), laminarin (La-OS), pustulan (Pu-OS), xylan (Xyl-OS), arabinoxylan (AraXyl-OS), arabinan (Ara-OS), arabinogalactan (AraGal-OS), mannan (Man-OS), glucomannan (GlcMan-OS), and galactomannan (GalMan-OS) [17]. Reaction rates with acceptors varied in the following order: MLG-OS > Cello-OS > Pu-OS > AraXyl-OS > La-OS > Xyl-OS > GlcMan-OS > Ara-OS. Minimal activities were seen with AraGal-OS, Man-OS, and GalMan-OS. Other factors influencing the activity of TmXET6.3 were DPs of Cello-OS or the positions of (1,4)-β- and (1,3)-β-linkages in MLG-OS [17]. TmXET6.3 could not catalyse the transfer of XG or HEC fragments on ionic (charged) acceptors.

As predicted from the alignments of TmXET6.3 with other XETs [17], including barley HvXET3, HvXET4, and HvXET6 isoforms [15,45,46], which showed the presence of residues identified in TmXET6.3 and were responsible for a broad acceptor specificity [17], all these XETs were able to catalyse the transfer of XG or HEC fragments to a wide panel of neutral acceptors [18]. As expected, there were only small differences in the hetero-transglycosylation efficacies between these isoforms, probably due to a residue variation in the acceptor binding sites. However, unlike TmXET6.3, the barley isoforms catalysed a novel type of hetero-transglycosylation reaction with negatively charged oligosaccharide acceptors, i.e., they catalysed the reaction between XG, cellulose, and the penta-galacturonide acceptor (fragment of a linear part of pectin) [18].

In XET sequences, enzymes could have either the Q102/R116 or H102/Q116 residue combinations (numbering of residues according to PttXET16A). The first combination is dominant and considered to be ancestral [10]. The Q108 residue of TmXET6.3 and the matching residues in non-specific barley HvXET isoforms corresponded to R116 in PttXET16A and are considered a signature residue for the XTH clade II [23]. Similarly, the H94 residue in TmXET6.3 and barley isoforms corresponded to Q102 in PttXET16A and is regarded as an additional signature residue of this XTH clade [10]. The shift from Q102/R116 (signature residues of the XTH clade I) to the H102/Q116 combination occurred at least five times during evolution and led to the convergent co-evolution of these residues, with the last event leading to the XTH clade II origin. Members of this clade, such as TmXET6.3 [17] and EfHTG [41,47], have broad substrate specificity, which can be considered an evolutionary advantage induced by the co-evolution of residues binding different saccharides [10].

The key contributions to the clarification of substrate specificity in XETs were made through experimental measurements combined with computational investigations and bioinformatics. This was possible due to the structural knowledge resulting from the atomic structure of PttXET16A [35]. Among the most important tools are the descriptions of the recognition mechanisms of substrate-enzyme complexes that are important for enzyme design. One example of a computational approach is the exploration of the dimeric XG nonasaccharide binding using molecular dynamics (MD) simulations [24], where one of XG nonasaccharides occupied the donor site creating a stable intermediate with an enzyme while the second XG nonasaccharide occupied the acceptor site. In both PttXET16A and TmNXG1 (which is XEH; EC 3.2.1.151), the Glc moiety of the nonreducing end of the XG nonasaccharide was located closest to the catalytic residues (which occupied the donor binding site) and altered its low-energy ^4^C_1_ into the ^1^S_3_ skew-boat conformation at the beginning of the MD simulation and maintained it [24].

Further, the benefit of computational methods can be illustrated by the fact that the substrate promiscuity in XETs from *Poaceae* was predicted by the molecular modelling of the GH16 family [48] before the first experimental evidence of broad substrate specificity in XETs was obtained [13,14]. Later, the specifics of these XG-OS interactions in the acceptor binding sites of several barley isoforms were demonstrated computationally and through enzyme kinetics [45]. The next valuable contribution was brought by the molecular modelling of HTG, where it revealed the residues that were responsible for the distinct substrate specificity of HTG [41]. It was shown that P10 and S34 participated in the donor substrate binding while L245 bound the acceptor substrate. It was notable that in other XETs, P10 is substituted by tryptophan and S34 by glycine residues [39]. Barley XET5 [14] and AtXTH3 [38], which exhibit high catalytic rates using cellulose as the donor, also contained in equivalent positions proline and serine residues as HTG, indicating the validity of this rationale. It was also suggested that the R246L mutation in HTG underlies the differences in Cello-OS binding [39]. However, this has yet to be verified because TmXET6.3 [17], EfXTHs [39], and barley HvXET3, HvXET4, and HvXET6 isoforms [18] catalysing transfers with Cello-OS have Arg in the equivalent positions, while AtXTH3 [38] has the R246K variation.

An additional contribution of the joined efforts of computational chemistry and bioinformatics provided fundamental information on the residues responsible for the differences in the acceptor specificity of the XTH clade I and II enzymes [17]. The signature residues mentioned above, specifically H94 and Q108, and certain lysine residues at the C-terminal end of TmXET6.3, such as K234, and K237, were identified [17]. Further, the residues responsible for the differences in substrate specificity amongst the XTH clade II members were identified, e.g., in certain non-specific barley XET isoforms that transferred XG or cellulose fragments onto charged acceptors [18], but not in others such as TmXET6.3 [17]. The H75 and R110 residues in barley HvXET3 and HvXET4 were identified to be responsible for these novel acceptor substrate specificities. In both cases, the accuracy of theoretical findings was verified and confirmed by mutational analyses [17,18]. 

In this work, the acceptor substrate specificities of specific PttXET16A and non-specific TmXET6.3 were studied using computational chemistry tools. The complexes of enzymes with the donor XG heptaoligosaccharide (XG-OS7) and a variety of acceptor substrates were obtained by molecular docking followed by MD simulations and combined with binding free energy calculations. We found that the stabilities of enzyme-substrate complexes were broadly in agreement with the experimental activity assays. Here, the instability of certain acceptors in the active site of PttXET16A was observed, while all tested oligosaccharide substrates were stable in the acceptor binding site of TmXET6.3. These findings are situated in the context of reaction mechanisms of specific and non-specific XETs and their functional roles during biogenesis and re-structuring of plant CWs. 

## 2. Results and Discussion

In this work, we used molecular docking with the induced fit docking protocol [49,50] and MD simulations combined with binding free energy calculations [51,52,53,54,55,56] to extract structural and thermodynamic features of XET-donor/acceptor substrate complexes. Protein-ligand interactions are classically examined by molecular docking, while MD simulations provide findings on protein conformational changes as proteins interact with docked ligands. We focused on PttXET16A [25,35] and TmXET6.3 [11,17], for which donor/acceptor substrate specificities are documented. 

### 2.1. Molecular Docking of Acceptor Substrates in TmXET6.3 and PttXET16A

A series of enzyme-donor/acceptor substrate complexes obtained by molecular docking were evaluated based on docking score parameters (the lowest docking score parameters correspond to the most favourable position) (Figure 1). The XG-OS acceptors (Table 1) docked in the acceptor site of PttXET16A exhibited a decreasing trend in docking score parameters with the increasing galactosylation of the acceptor side XG chains [XG-OS7 (−6.79 kcal/mol) > XG-OS8 (−7.01 kcal/mol) > XG-OS9 (−7.96 kcal/mol)], while in TmXET6.3, XG-OS8 had the lowest score of −6.86 kcal/mol, followed by XG-OS7 (−6.71 kcal/mol), and XG-OS9 (−6.54 kcal/mol). The respective scores for La-OS4 and Pu-OS4 were lower for TmXET6.3 (−6.02 kcal/mol and −6.93 kcal/mol) compared to PttXET16A (−5.46 kcal/mol and -−5.35 kcal/mol). However, the respective scores with Cello-OS4, Man-OS4, and Xyl-OS4 were lower when these acceptors were docked in the acceptor site of PttXET16A (−4.66 kcal/mol, −6.78 kcal/mol, and −3.52 kcal/mol) rather than in the active site of TmXET6.3 (−3.50 kcal/mol, −6.05 kcal/mol and −2.20 kcal/mol). This finding disputed the experimentally determined activities of TmXET6.3 using these acceptors [17]. Here, the highest activity was observed with Pu-OS, followed by Cello-OS, while the lowest activity was observed with Man-OS, which had the second-best docking score parameter. The activities of PttXET16A with the latter acceptor substrates were not reported [25,35].

Differences in docking score parameters between various GlcMan-OS indicated the importance of side chain position during the binding of acceptors. GlcMan-OS3 with a side chain located on the third saccharide moiety from the nonreducing end had the lowest score of −7.17 kcal/mol, followed by GlcMan-OS4 (−6.0 kcal/mol) with a substituted reducing end moiety, and GlcMan-OS2 (−5.59 kcal/mol), whereas the highest score was observed with GlcMan-OS1 (−5.25 kcal/mol), caused by a side chain branching from the saccharide moiety at the nonreducing end, and resulting from a larger size of the Glc (hexose) side chain moiety, compared to that of xylose (pentose) occupying the same position in the XG-OS substrates. AraXyl-OS, with a second xylose moiety from the nonreducing end substituted by arabinose, had a higher score (−4.95 kcal/mol) compared to all GlcMan-OS acceptors but lower than Xyl-OS. The docking score parameter values of the MLG-OS acceptors [containing (1,4;1,3)-β-linked Glc moieties] signposted the significance of the (1,3)-β-glycosidic linkage locations among (1,4)-β-linked residues, which caused a kink in backbones. Among MLG acceptors, the lowest score was achieved with MLG-OSC (−6.82 kcal/mol) with the (1,3)-β-linkage located closest to the reducing end, followed by MLG-OSA (−6.53 kcal/mol) with the (1,3)-β-linked nonreducing end, while the highest docking score parameter was found with MLG-OSB (−5.52 kcal/mol).

Considering the observed conflicting results obtained by docking and activity assays, MD simulations with a length of up to 1000 ns were performed to test the stability of selected XET-donor/acceptor substrate complexes and to explore the interactions between individual saccharide-binding amino acid residues and acceptor substrates. Based on acquired trajectories, the time dependence of distances between the atoms participating in the glycosidic linkages between the donor and acceptor substrates was evaluated, i.e., between the C1 atom of the reducing saccharide moiety of the XG-OS7 donor and the O4 atom of the nonreducing end saccharide moiety of acceptors. The generated XET-donor/acceptor substrate complexes were further characterised by calculating their binding free energies from the last 200 ns of MD simulations. 

### 2.2. The Stability of Enzyme-Donor/Acceptor Complexes with XG Oligosaccharide Acceptors

It has been established that XG and XG-OS are the key saccharide molecules serving as donor and acceptor substrates for XETs [1,2,3,4,5,6,7,8,12,13,14,15,16,17,18,19]. Here, the experimentally determined reaction rates of XETs depend on the degree of the galactosylation of xylose side chain residues in XG-OS [18]. Further, the acceptor preference of non-specific barley XET isoforms varies with XG-OS7, XG-OS8, and XG-OS9 [18], while the preference of PttXET16A for these acceptors remains to be defined [25,35]. In the case of the barley HvXET isoforms, the differences in the XG-OS9 interactions in the acceptor sites were studied computationally and compared with the experimental enzyme kinetics data [45]. 

In this work, we report the stability of enzyme-donor/acceptor complexes with the XG-OS7, XG-OS8, and XG-OS9 acceptor substrates in binding sites of TmXET6.3 and PttXET16A and compare these and experimental data. During the first 640 ns of simulation, XG-OS7 (Figure 2; RMSD values in Appendix A) in the acceptor-binding site of TmXET6.3 had an average distance from the donor at 5.16 Å (3.15-8.47 Å). However, a change in side chain position allowed the movement of the main chain closer to the donor substrate, as the acceptor was stabilised to an average distance of 4.40 Å (2.92–6.62 Å). Conversely, XG-OS7 in the active site of PttXET16A remained relatively stable during the whole MD simulation time, with an average distance from the donor of 5.49 Å (3.03–7.74 Å). The positions of XG-OS8 were stable during entire simulations with both enzymes (Figure 2; RMSD values in Appendix A.) with an average distance of 5.82 Å (3.50–8.84 Å) in TmXET6.3 and 5.68 Å (3.00–9.65 Å) in PttXET16A. Similarly, XG-OS9 (Figure 2; RMSD values in Appendix A) had an average distance from the donor of 5.17 Å (3.39–7.68 Å) in TmXET6.3 and 5.24 Å (3.07–7.73 Å) in PttXET16A. Fluctuations were greater when XG-OS8 served as an acceptor compared to XG-OS9, especially in the active site of PttXET16A. This difference could be the consequence of decreased mobility of XG-OS9 caused by the presence of another galactose moiety on one of the side chains.

During MD simulations with TmXET6.3 and PttXET16A, the side chains binding the second and third Glc moiety from the nonreducing end moved towards the loop containing Y230 and Y250 (Appendix A). The xylose moiety of the second side chain from the nonreducing end interacted with multiple residues in TmXET6.3 and PttXET16A (Table 2; interactions in Appendix A), and notably, these interactions occurred through the residues in the same positions in the tertiary structures or models. The only difference was the interaction of R258, which was observed only in PttXET16-34A. The side chains of all XG-OS acceptors interacted with H94, Q102, and Q108 of TmXET6.3 and R116 in PttXET16-34A (with the exception of XG-O9 in PttXET16A), which are considered to be the signature residues of XTH clades I and II [10].

Generally, the number of interactions between saccharide moieties of the backbone acceptor chain decreased from the nonreducing to reducing ends, suggesting that a more significant role in acceptor binding is played by the saccharide moieties located closer to the location in the catalytic site, where a new glycosidic bond is formed. However, the interactions between G175 (TmXET6.3) and G183 (PttXET16-34A) and the third and fourth saccharide moieties could also contribute to the stabilisation of the main chain in both enzymes.

The decreasing trend in binding free energies illustrated the increase in the acceptor stability with the size of side chains (Table 3). Although the differences were more striking between XG-OS7 (−41.36 kcal/mol) and XG-OS8 (−65.99 kcal/mol) in TmXET6.3, in PttXET16A these differences were similar for XG-OS7 (−46.12 kcal/mol) and XG-OS8 (−46.20 kcal/mol), but the binding free energy decreased significantly with XG-OS9 (−74.99 kcal/mol). Overall, the binding free energies of XG-OS7 and XG-OS9 were similar between the two enzymes, while XG-OS8 had significantly lower energy in TmXET6.3. These data indicated a strong preference of XG-OS9 for PttXET16A, although this needs to be experimentally confirmed. Alternatively, in the case of TmXET6.3, the binding free energy for XG-OS8 and XG-OS9 were similar and corresponded to the experimentally determined activities, while the reaction rates were significantly lower with XG-OS7 (B.S., unpublished).

### 2.3. The Stability of Enzyme-Donor/Acceptor Complexes with Linear Oligosaccharide Acceptors

Cello-OS4, La-OS4, Pu-OS4, Man-OS4, and Xyl-OS4 were all stable in the active site of non-specific TmXET6.3, while in the active site of specific PttXET16A, the only stable linear acceptors were Cello-OS4 and La-OS4. The lower fluctuations in distances between donors and acceptors compared to XG-OS (7–9) likely resulted from their limited movements due to the side chain mobilities of XG-OS (7–9). The position of Cello-OS4 in the active site of TmXET6.3 was considerably more stable during the whole duration of the MD simulation (Figure 3; RMSD values in Appendix A), with an average distance of 4.97 Å (3.59–7.68 Å) from the donor. Unlike XG-OS (7–9), the interaction of Cello-OS4 with Q108 in TmXET6.3 was not engaged (Table 4; interactions in Appendix A) since the chain was unable to move close enough to this residue, while with XG-OS (8–9) the interactions with Q108 occurred with side chains of XG-OS (8–9). Cello-OS4 in the active site of PttXET16A (Figure 3; RMSD values in Appendix A) was relatively stable during the duration of the MD simulation with an average distance of 5.01 Å (3.06–12.95 Å). The chain of Cello-OS4 during simulation changed its position, approaching the loop containing S257 (Figure 4). TmXET6.3 has L237 in this position, whose chain is longer than that of serine, and thus, it could act as a steric barrier preventing similar changes in the positions of saccharide moieties in the active site of TmXET6.3. The ultimate difference between the positions of Cello-OS4 in PttXET16-34A and TmXET6.3 allowed the interaction between Cello-OS4 and R116 only in the PttXET16-34A complex (Table 5; interactions in Appendix A). 

The position of La-OS4 was relatively stable in the active site of TmXET6.3 during the first 350 ns (Figure 3; RMSD values in Appendix A) with an average distance of 5.84 Å (2.99–8.96 Å) from the donor, followed by a lessening of that distance to an average of 4.23 Å (2.86–7.01 Å) during 350–650 ns, and finally reaching a stable position at an average distance of 5.47 Å (3.48–7.90 Å) after 650–1000 ns of MD simulations. However, La-OS4 in the active site of PttXET16A (Figure 3; RMSD values in Appendix A) remained stable during the duration of the MD simulation, with an average distance of 4.64 Å (3.11–6.93 Å) between the O4 atoms of the acceptor and the C1 atoms of the donor. Similar to Cello-OS4 and La-OS4 in TmXET6.3, the number of interactions between the acceptor and the enzyme exhibited a decreasing trend from the nonreducing towards the reducing ends, with the reducing end moiety lacking any interaction at over 50% of MD simulation times (Table 4; interactions in Appendix A). The position of La-OS4 at the end of the MD simulation was comparable to that of the Cello-OS4 acceptor (Appendix A).

A larger oscillation in the distance between the O4 atom of the Pu-OS4 acceptor and the C1 atom of the XG-OS7 donor occurred in the active site of TmXET6.3 (Figure 3; RMSD values in Appendix A). During the first 650 ns, this distance oscillated between 2.89–13.75 Å with an average of 5.99 Å. Subsequently, during 650-850 ns, the acceptor moved to an average distance of 7.57 Å (4.02–11.07 Å), and towards the end of the MD simulation, the acceptor stabilised at an average distance of 5.56 Å (3.48–10.47 Å) over the last 150 ns of the MD simulation time. The difference between the stability and position (Appendix A) of Pu-OS4 and other acceptors likely resulted from a difference in its structure caused by a (1,6)-β-linkage interconnecting Glc residues. Unlike the case of Cello-OS4, the stable position of Pu-OS4 allowed the interactions between the third saccharide moiety from the nonreducing end with W171, D170, and G175 of TmXET6.3 (Table 4; interactions in Appendix A) combined with the W108 interaction with the reducing end Glc moiety. This Pu-OS4 acceptor and the XG-OS (7-9), Cello-OS4, and La-OS4 acceptors displayed a decreasing trend in numerous interactions between the acceptor and the enzyme, starting from the nonreducing towards the reducing ends. In the active site of PttXET16A, Pu-OS4 remained stable during the first 320 ns (Figure 3; RMSD in Appendix A) with an average distance of 6.26 Å (3.09–13.61 Å) from the donor, however, this 320 ns period of time was followed by destabilisation, as the distance between the O4 atom of Pu-OS4 and the C1 atom of the donor increased to up an average of 9.84 Å (6.62–17.33 Å). This increase in the distance resulted in the formation of only one interaction between Pu-OS4 and PttXET16A that occurred during more than 50% of MD simulation times (Table 4; interactions in Appendix A), where Pu-OS4 also lacked any interactions with the XG-OS7 donor.

The binding free energy of the La-OS4 acceptor (Table 5) was relatively low for both enzymes, further illustrating the stability of this acceptor in the active sites of both enzymes. In PttXET16A (−41.06 kcal/mol), the binding free energy value was slightly higher than in TmXET6.3 (−43.50 kcal/mol). Notably, the binding free energy of Cello-OS4 (−26.5 kcal/mol) was relatively high in PttXET16A, despite apparent stability during the entire MD simulation time. The binding free energy of TmXET6.3 in complex with Pu-OS4 (−45.17 kcal/mol) was comparable to that of Cello-OS4 (−48.45 kcal/mol), while the value for Pu-OS4 in PttXET16A (−32.62 kcal/mol) was higher compared to that of TmXET6.3. In contrast to the XG-OS and Cello-OS4 acceptors, which contain (1,4)-β-glycosidic linkages in main chains (Table 1), the (1,6)-β-linkage in Pu-OS4 caused a significant geometry alteration of the main chain, thus, destabilising the binding of Pu-OS in PttXET16A.

Compared to La-OS4 and Pu-OS4, Man-OS4 and Xyl-OS4 displayed fewer oscillations in the distances from the donor in TmXET6.3. Man-OS4 was stabilised in a similar position as the Cello-OS4 acceptor (Figure 5) and remained stable during the whole MD simulation time with an average distance of 4.94 Å (3.29–13.20 Å) from the donor (Figure 3; RMSD values in Appendix A). The previously observed decreasing trend in some interactions between the enzyme and the acceptor (from the nonreducing to reducing ends) also occurred with the Man-OS4 acceptor (Table 4; interactions in Appendix A). On the other hand, in PttXET16A, the Man-OS4 acceptor was destabilised after less than 50 ns of simulation time (Figure 3; RMSD values in Appendix A), and the distance between the acceptor and the donor increased up to an average of 18.42 Å (2.91–28.07 Å). The analysis of the interactions between this acceptor and the PttXET16A residues showed a lack of any interactions above 50% of the MD simulation time with the first and second saccharide moieties from the nonreducing end (Table 4; interactions in Appendix A). The increased number of interactions compared to other acceptors with the third and fourth mannose moieties resulted in incorrect positioning of the Man-OS4 acceptor.

The Xyl-OS4 acceptor remained stable during the entire MD simulation in the active site of TmXET6.3 [11], with an average distance from the donor of 4.93 Å (3.52–9.16 Å) (Figure 3; RMSD values in Appendix A). Similar to other acceptors, Xyl-OS4 displayed a declining trend in the number of interactions between the acceptor and the enzyme from the nonreducing to the reducing ends (Table 4; interactions in Appendix A). The distance of Xyl-OS4 from the donor was relatively stable in PttXET16A during the first 450 ns (Figure 3; RMSD values in Appendix A), with an average distance of 5.97 Å (3.60–9.76 Å), whereas after this time, this distance reached an average distance of 9.70 Å [11] (3.37–17.12 Å) (*cf*. Supplementary Video). The Xyl-OS4 acceptor also formed only two interactions with PttXET16A (Table 4; interactions in Appendix A). Similar to Pu-OS4, both acceptors lacked reliable interactions with the donors, unlike TmXET6.3, which exhibited these interactions with all acceptors. These data are in good agreement with the predicted specificity of PttXET16-34A, and thus this enzyme should not mediate the hetero-transglycosylation reactions with Man-OS4 and Xyl-OS4.

The calculated value of the binding free energy of Man-OS4 (Table 5) was lower in PttXET16A (−38.26 kcal/mol) compared to that of TmXET6.3 (−32.15 kcal/mol). However, this observation in PttXET16A resulted from a pose that was adopted during the last 200 ns—although this pose was stable, it was inappropriate for the glycosidic bond formation between the donor and acceptor substrates. Following the activity assays, the value of the binding free energy of Man-OS4 in TmXET6.3 was higher compared to that of Pu-OS4, where either the presence of a (1,6)-β-linkage or the substitution of Glc moieties for mannose led to acceptor instabilities in PttXET16A. 

In the case of the Xyl-OS4 acceptor, which consists of xylose moieties, it was unstable in PttXET16A (similar to Man-OS4 and Pu-OS4), also exhibiting a higher binding free energy (−23.38 kcal/mol) compared to that of TmXET6.3 (−31.84 kcal/mol). The binding free energy for TmXET6.3 with Xyl-OS4 agreed with the experimental data, where we observed a lower activity compared to those with the Cello-OS4 and Pu-OS4 acceptors [17].

### 2.4. Effects of Linkage Positions in Main Chains of Mixed-Linkage Glucan Acceptors on the Stability of Enzyme-Donor/Acceptor Complexes

MD simulations of three TmXET6.3-XG-OS7/MLG-OS4 (A–C) complexes, wherein the acceptors differed in the positions of (1,3)-β-linkages, illustrated the significance of the (1,3)-β-linkage placements among (1,4)-linked Glc moieties. The positions of MLG-OSA with the (1,3)-β-linkage located closest to the nonreducing end Glc moiety were highly unstable during the beginning of the simulations, with an average distance of 12.66 Å (4.49–30.33 Å) from the donor (Figure 6; RMSD values in Appendix A). The instability of the MLG-OSA position was also obvious by a lack of interactions with the residues of TmXET6.3 taking place after longer than 50% of the MD simulation time (Table 6). The TmXET6.3-XG-OS7/MLG-OSB complex with the (1,3)-β-linkage of the acceptor between the second and third saccharide moiety was slightly unstable during the first 320 ns of simulation (Figure 6; RMSD values in Appendix A) with an average distance of 7.46 Å (3.38–14.45 Å). However, after this time, the stabilisation of its position occurred, and the acceptor remained at an average distance of 5.47 Å (3.33–8.13 Å) from the donor. MLG-OSC with the (1,3)-β-linkage present between the third and fourth moiety appeared to be the most stable (Figure 6; RMSD values in Appendix A) among all tested MLG-OS (A–C) acceptors, with an average distance of 4.99 Å (2.89–9.00 Å), between the O4 atom of the acceptor and the C1 atom of the donor. MLG-OSB and MLG-OSC (Table 6; interactions in Appendix A) displayed a similar number of interactions as those observed with Cello-OS4 and La-OS4. The only notable difference when comparing these two acceptors was the interaction of Y230 with the reducing end moiety of MLG-OSB. The positions of MLG-OSB and MLG-OSC (Figure 7) at the end of the MD simulation were strikingly similar to those of Cello-OS4 and Xyl-OS4—the difference between these two acceptors was in the interactions of the reducing end moiety of MLG-OSB with Y230, and the interactions of the MLG-OSC nonreducing end moiety with D79.

Further exploration of the instability of binding of the MLG-OS4 (A–C) acceptors in TmXET6.3 showed that the value of the binding free energy of MLG-OSA (−33.34 kcal/mol) was significantly higher compared to MLG-OSB and MLG-OSC (Table 7). The binding free energies of MLG-OSB (−42.62 kcal/mol) and MLG-OSC (−41.64 kcal/mol) were comparable, while the MLG-OSB acceptor, despite the oscillations in its position at the beginning of the MD simulation, had slightly lower energy compared to that of MLG-OSC. The noteworthy fact was that the MLG-OSB acceptor had the lowest binding free energy (that corresponded to the activity assays [17]), which was higher compared to the energies of other MLG-OS acceptors.

*Effect of side chain positions on the stability of enzyme-donor/acceptor complexes:* To determine the effect of side chain position on the TmXET6.3-donor/acceptor complexes, MD simulations were performed with GlcMan-OS (identical main chain as Man-OS4 but there were differences in the side chains) and AraXyl-OS (the second xylose moiety from the nonreducing end was substituted). GlcMan-OS1 with a side chain positioned at the nonreducing end was destabilised, shortly after the beginning of the MD simulation (Figure 8; RMSD values in Appendix A), with an average distance from the donor of 8.73 Å (3.76–17.68 Å) thus it was unfit for bond formation. Conversely, GlcMan-OS2 with the substituted second saccharide moiety from the nonreducing end was relatively stable during the MD simulation, with an average distance between C1 of the donor and O4 of the acceptor of 5.0 Å (3.01–7.60 Å). GlcMan-OS3 and GlcMan-OS4 substituted on the third and fourth moieties were also stable. An average distance from the donor for GlcMan-OS3 was 5.78 Å (3.19–14.97 Å) and 6.12 Å (3.41–12.71 Å) for GlcMan-OS4. AraXyl-OS, during the first 200 ns of MD simulations (Appendix A), bound stably with an average distance from the XG-OS7 donor of 4.09 Å (2.95–6.65 Å). However, in the 200–550 ns time frame, this acceptor moved marginally away from the donor to the distance of 5.57 Å (2.93–14.82 Å). Higher fluctuations were observed during 550–720 ns with an average distance of 6.77 Å (3.15–12.92 Å), and following this time interval, the position of AraXyl-OS stabilised with an average distance of 5.53 Å (4.30–8.53 Å).

In TmXET6.3, the interaction between D170 and GlcMan-OS1 (Table 8; interactions in Appendix A) resulted from an unsuitable position of the acceptor (Figure 9), while with GlcMan-OS2, the side chain interacted with Q108, G175, and W171 (Table 8; interactions in Appendix A), and the Glc moiety of GlcMan-OS3 featured interactions with W230, R238, D235, and K237 (Table 8; interactions in Appendix A). Finally, the GlcMan-OS4 acceptor showed no interactions between side chains and TmXET6.3 active site residues (Table 8; interactions in Appendix A).

The interactions of AraXyl-OS with TmXET6.3 (Appendix A) were similar to those of Xyl-OS4, with the only notable difference being the interaction between the nonreducing end moiety of AraXyl-OS with Q108 and the lack of any interactions with G175. The side chain that distinguishes AraXyl-OS from Xyl-OS did not show any interaction that lasted beyond 50% of the MD simulation time. 

The calculated binding free energy values (Table 9) confirmed that GlcMan-OS2 was the most stable acceptor in the active site of TmXET6.3 (−47.28 kcal/mol), followed by GlcMan-OS4 (−42.75 kcal/mol), and GlcMan-OS3 (−32.3 kcal/mol). In agreement with the higher experimentally determined activity of TmXET6.3 with GlcMan-OS than with Man-OS [17], the binding free energy values were lower for all GlcMan-OS acceptors including Man-OS4 (−32.15 kcal/mol), except for GlcMan-OS1 (−28.45 kcal/mol). As experimental assays use a mixture of variously substituted GlcMan-OS, these data illustrate the preference of TmXET6.3 to substrates with a defined position of side chains. The instability of GlcMan-OS1 also pointed to the difference between acceptors with various types of side chains positioned at the moieties of the nonreducing ends since XG-OS bonded stably during the examined MD simulation time. The binding free energy of AraXyl-OS (−45.32 kcal/mol) was comparable to that of GlcMan-OS2 (−47.28 kcal/mol) and GlcMan-OS4 (−42.75 kcal/mol), which was significantly lower compared to Xyl-OS4 (−31.84 kcal/mol), thus ascertaining the role of side chains during the stabilisation of this acceptor. This finding is in good agreement with the higher activity of TmXET6.3 with the AraXyl-OS acceptor than with Xyl-OS that were experimentally observed [17].

## 3. Experimental Section

### 3.1. Homology Modelling of TmXET6.3

The homology model of TmXET6.3 (the nucleotide sequence of TmXET6.3 is available in GenBank under HF968473 and the protein sequence in UniprotKB under V5ZEF7) was constructed [17] based on the coordinates of the crystal structure of hybrid aspen PttXET16A as the template (Protein Data Bank 1UN1) [35]. Sequence identity of TmXET6.3 to PttXET16A is 41.5% [57]. Homology models were generated in Modeller9v6 [58]. Five models were generated, with refined loops for each model. The best-scoring TmXET6.3 model was chosen based on the lowest probability density function and the lowest energy of refined loops. The root-mean-square-deviation (RMSD) value of the Cα residues between TmXET6.3 and template structure was calculated using the *cpptraj* utility of AmberTools14 and corresponds to 0.23 Å [17,51]. The structural model of TmXET6.3 is available in the Protein Model DataBase under PM0081526.

### 3.2. Generation of XET-Donor/Acceptor Complexes

*Donor substrate:* The coordinates of the XG heptaoligosaccharide (XG-OS7) donor (structural formula in Table 1) were those of the XG nonaoligosaccharide (XG-OS9), taken from the crystal structure of the TmNXG1-DYNIIG mutant complex [Protein Data Bank (PDB) 2VH9] [24], after removing two terminal galactose moieties from xylopyranosyl residues.

*Acceptor substrates:* The structures of acceptors (formulae in Table 1) were created using the tools available on the GLYCAM-web server [59], followed by preparation for molecular docking using the LigPrep tool [60]. Protein structures of TmXET6.3 and PttXET16A were prepared for docking using Protein Preparation Wizard [61]. A series of acceptors were docked into the active sites of TmXET6.3 and PttXET16A with the Glide program (Glide; version 6.7; Schrödinger LLC: New York, NY, USA, 2015) [49], using the Extra Precision protocol and a box size of 14 Å × 14 Å × 14 Å (used for the ligand). The acceptor main chains were docked using the XG-OS6 ligand from the PttXET16A complex (PDB 1UMZ) with core constraints and side chain flexibility. The resulting structure of the acceptor substrate was re-docked using the induced fit docking protocol (Induced Fit Docking Protocol 2015-2; Glide version 6.4, Prime version 3.7; Schrödinger LLC: New York, NY, USA, 2015) [50]. The flexibility of protein residues was allowed at a distance of up to 7 Å from acceptors. The best-scoring poses of enzyme-donor/acceptor complexes for each donor/acceptor combination were selected based on the docking score parameters and stability of the complexes that were verified using MD simulations. 

### 3.3. Molecular Dynamics (MD) Simulations

The structures of enzyme-donor/acceptor complexes obtained from molecular docking were prepared for MD simulations in *tleap*, which is part of the Amber16 [51] program package. The preparation included the application of Amber ff99SB force field parameters for protein and GLYCAM06 [52] parameters for donor and acceptor substrates and solvation with TIP3P water molecules [53] in a cubic box with a 15 Å water layer height. In the case of PttXET16A, four Cl^−^ ions were added to neutralise charges. The resultant structures were optimised and equilibrated under limited periodic conditions. During the first step of the equilibration, the positions of water molecules were optimised using the steepest descent method, and protein/saccharide molecules were restrained using the 50 kcal/mol harmonic potential. The next step included the heating of the system to 300 K during a 100 ps NVE simulation (under conditions of constant energy, volume, and number of particles) followed by a 300 ps NPT simulation (under conditions of constant temperature, pressure, and number of particles) with 1 bar pressure. During the following 10 ps steps, the harmonic potential applied to the complexes was gradually reduced to 25 and 10 kcal/mol, following the 50 ps simulation with a harmonic potential of 5 kcal/mol and the 70 ps simulation with 2.5 kcal/mol. In the next step, we applied a 300 ps simulation with a harmonic potential reduced to 1 kcal/mol, and the last step consisted of a 300 ps simulation without a harmonic potential. Equilibrated structures were used for MD simulations using the CUDA implementation [54] of *pmemd*, which is part of the Amber 16 package. MD simulations were performed under limited periodic conditions using an NPT ensemble with 300 K temperature and 1 bar pressure. The simulation length was extended to 1000 ns, and the integration step was 2 fs. Hydrogen atoms were restrained using the SHAKE algorithm [55], and electrostatic forces were calculated using the Particle Mesh Ewald method [56], with snapshots saved every 5 ps for each trajectory. Resultant trajectories were used for analyses of interactions between amino acid residues and acceptor substrates and for calculations of binding free energies. Binding free energies were calculated from the last 200 ns of trajectories using the MMPBSA.py script, which is part of the AmberTools16 [51], where every third snapshot from each trajectory is considered.

### 3.4. Comparison of Theoretical and Experimental Data

The results of theoretical stabilities of enzyme-donor/acceptor complexes obtained by MD simulations were corroborated with the experimental data of PttXET16A [25,35] and TmXET6.3 [17], obtained in ‘in vitro’ enzyme assays with selected donor/acceptor substrates.

*Supplementary Video:* Visualisation of the dynamics of the XG-OS7 donor and Xyl-OS4 acceptor substrates bound in the active site of PttXET16A, obtained by MD simulation.

The visualisation of the MD trajectory of the XG-OS7 donor and Xyl-OS4 acceptor substrates in the active site of PttXET16A reveals the instability of the acceptor. After 20 ns the Xyl-OS4 chain changes its position and approaches the loop with S257. However, unlike Cello-OS4 this interaction does not stabilise the acceptor. After 50 ns the first signs of destabilisation of Xyl-OS4 can be observed, and after 460 ns its chain moves into a position that is unfavourable for glycosidic bond formation. Conversely, the Xyl-OS4 acceptor in the active site of TmXET6.3 remains stable during the duration of the MD simulation (*cf*. Figure 3).

## 4. Conclusions

In this work, we used molecular docking and MD simulations combined with binding free energy calculations to define the acceptor substrate binding specificity of two widely studied plant XETs—PttXET16A and TmXET6.3. Using these computational approaches, we conclude that linear acceptors with the Glc residues interconnected by (1,4)- and (1,3)-linkages are stable in the active sites of TmXET6.3 and PttXET16A enzymes. Conversely, the acceptors with (1,6)-linkages (Pu-OS4) and the acceptors with main chain moieties different from Glc, such as Man-OS and Xyl-OS, are only stable in the active site of non-specific TmXET6.3. From the evolutionary point of view, it is notable that PttXET16A clusters with other XETs similar to the barley XET5 isoform, with experimentally determined high activities with the XG/Cello-OS substrate pair. In TmXET6.3, the XG-OS7/MLG-OS complexes with the acceptors featuring (1,3)-β-linkages between Glc moieties varied in stability which agreed with experimental results; here the acceptors with the (1,3)-β-linkage between the third and fourth Glc moieties were the most stable. The simulations with GlcMan-OS acceptors with different side chain positions indicated the varying preference of TmXET6.3 towards these substrates and corroborated data obtained from experimental assays performed with a mixture of GlcMan-OS.

Although PttXET16A is currently considered to be a strictly specific XET enzyme, our observations suggest that PttXET16A could potentially catalyse transfer reactions with XG fragments to the acceptors composed of Glc residues connected by (1,4)- and (1,3)-linkages—although these activities so far remain experimentally unavailable. 

There is a surprising lack of information on the hetero-transglycosylation activities of structurally characterised XETs [34]. Another problem arises from the lack of information on XET enzymes (at the atomic level) other than PttXET16A, where molecular modelling of hetero-transglycosylating XETs could only use the template of substrate-specific PttXET16A [25,35]. Such a paucity of information may lead to errors when extracting and interpreting the structural information of modelled hetero-transglycosylating XETs [62]. Another improvement could be gained from using tailor-made force fields during the docking of substrates into the active sites of XETs [63]. 

In summary, our data contribute to the definition of substrate specificity in higher plant TmXET6.3 and PttXET16A XG-endotransglycosylases and dissect the structural and thermodynamic foundation of the exclusively carbohydrate-linked transglycosylation catalytic function of these enzymes [10,11,19]. Additionally, this study suggests that XET enzymes contribute not only to XG but also to other biopolymer integrations into plant CWs and potentially to the synthesis of the XG-hybrid molecules, a feature that underlies the complexity of plant CWs [64,65,66].

## Figures and Tables

**Figure 1 ijms-23-11838-f001:**
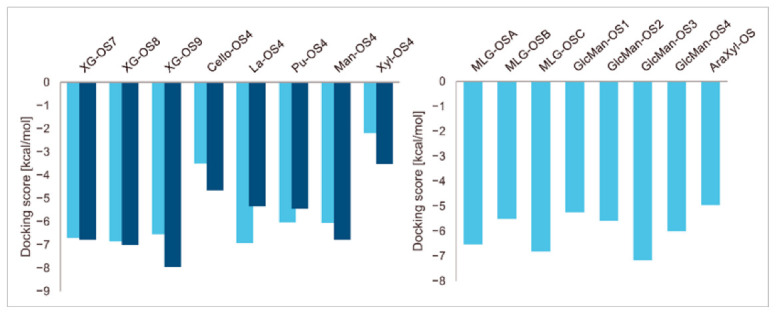
Induced fit docking score parameters of selected acceptor substrates docked into the active sites of TmXET6.3 (bright blue) and PttXET16A (dark blue).

**Figure 2 ijms-23-11838-f002:**
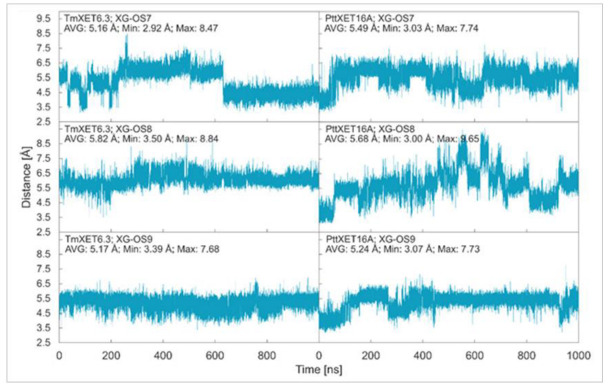
Time dependence of the distance (Å) between the C1 atom of the donors and the O4 atom of acceptors during MD simulations with TmXET6.3 and PttXET16-34A. The C1 and O4 atoms form a glycosidic bond.

**Figure 3 ijms-23-11838-f003:**
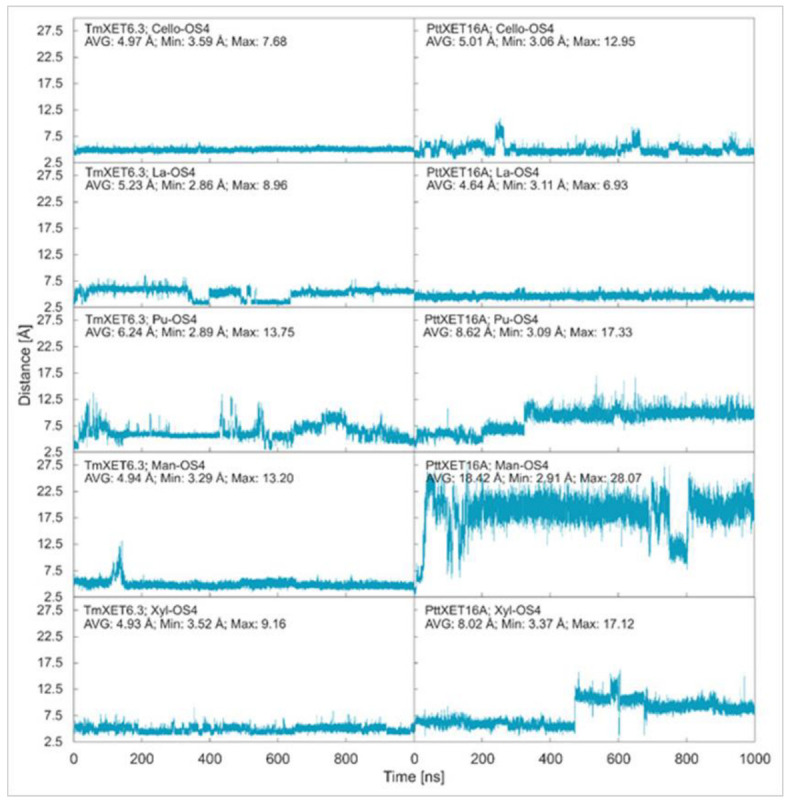
Time dependence of the distance (Å) between the C1 atom of the XG-OS7 donor and the O4 atoms of acceptors during MD simulations with TmXET6.3 and PttXET16-34A. The C1 and O4 atoms form a glycosidic bond.

**Figure 4 ijms-23-11838-f004:**
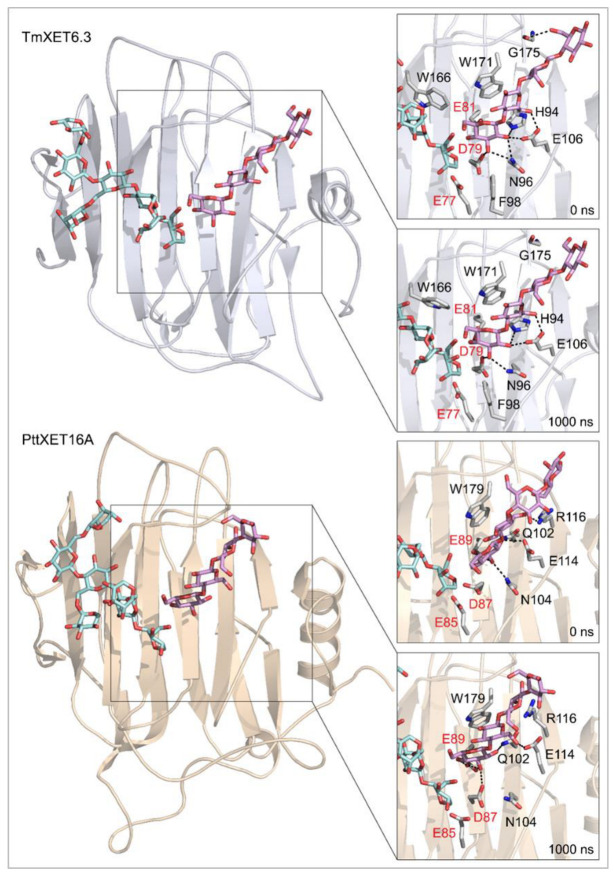
Positions of the XG-OS7 donor and docked Cello-OS4 acceptor substrates in TmXET6.3 or PttXET16A (left). The residues that interact with acceptors over 50% of MD simulation times at distances of up to 4.0 Å (right, black letters) shown at the beginning (0 ns) and after 1000 ns of MD simulation times. Catalytic residues are shown in red letters.

**Figure 5 ijms-23-11838-f005:**
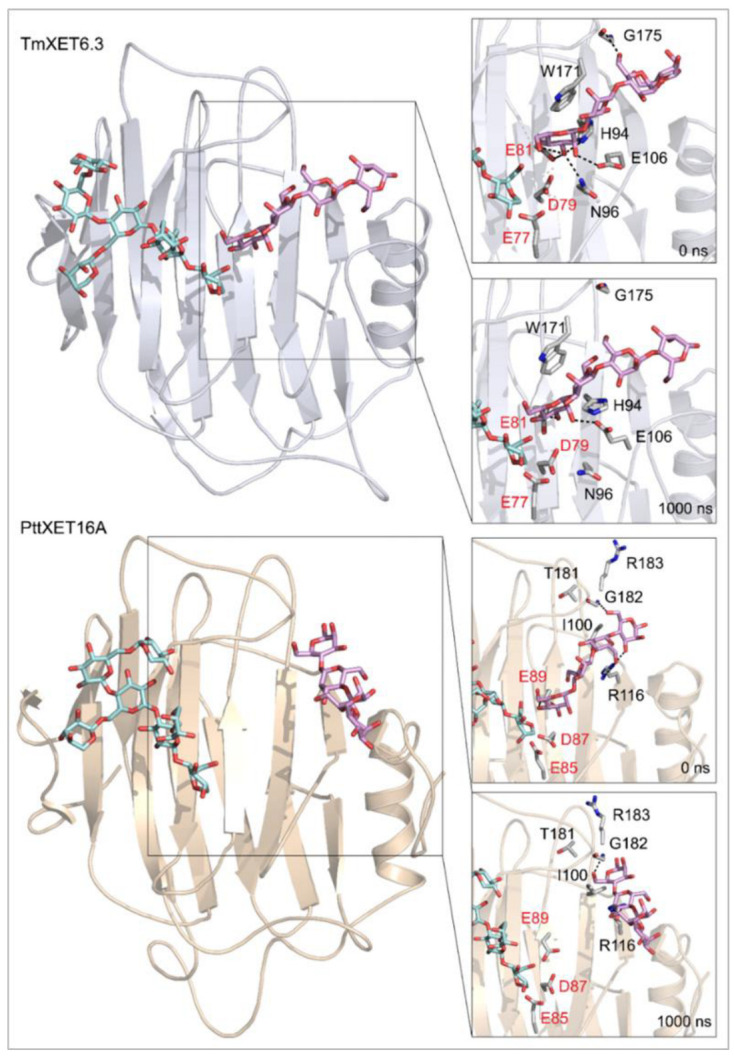
Positions of the XG-OS7 donor and docked Man-OS4 acceptor substrates in TmXET6.3 or PttXET16A (left). The residues that interact with acceptors over 50% of MD simulation times at distances of up to 4.0 Å (right, black letters) are shown at the beginning (0 ns) and after 1000 ns of MD simulation times. Catalytic residues are shown in red letters.

**Figure 6 ijms-23-11838-f006:**
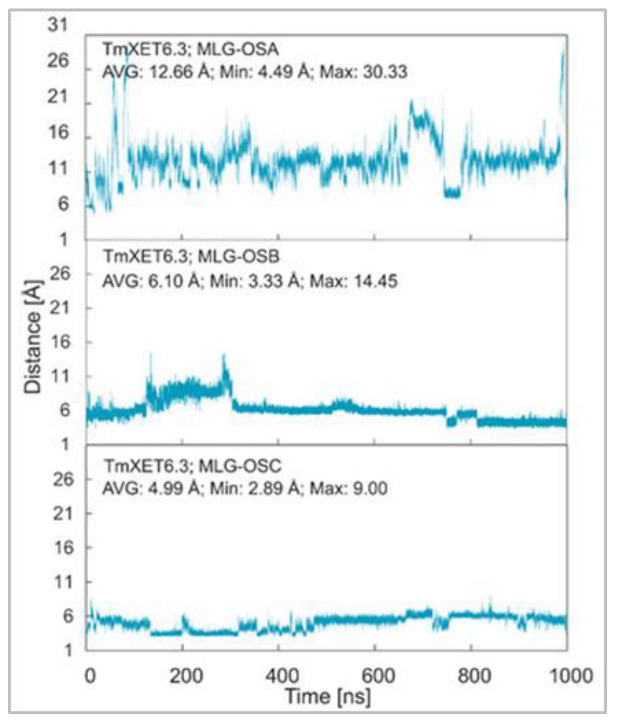
Time dependence of the distance (Å) between the C1 atom of the XG-OS7 donor and the O4 atom of the MLG-OS acceptors during MD simulations with TmXET6.3. The C1 and O4 atoms form a glycosidic bond.

**Figure 7 ijms-23-11838-f007:**
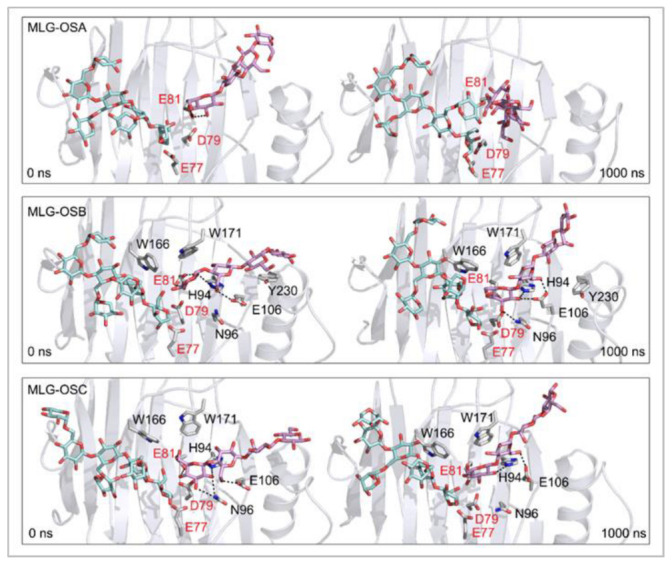
Positions of the XG-OS7 donor and docked MLG-OS acceptor substrates that interact with the TmXET6.3 residues (interactions shown at the acceptor sites). The residues that interact with acceptors over 50% of MD simulation times at distances of up to 4.0 Å (right, black letters) are shown at the beginning (0 ns) and after 1000 ns of MD simulation times. Catalytic residues are shown in red letters.

**Figure 8 ijms-23-11838-f008:**
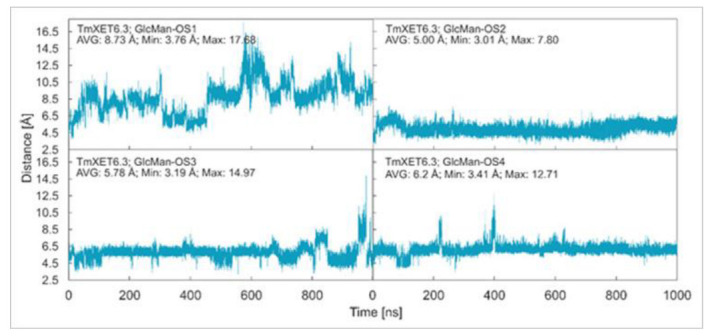
Time dependence of the distance (Å) between the C1 atom of the XG-OS7 donor and the O4 atom of the GlcMan-OS acceptor during MD simulations with TmXET6.3. The C1 and O4 atoms form a glycosidic bond.

**Figure 9 ijms-23-11838-f009:**
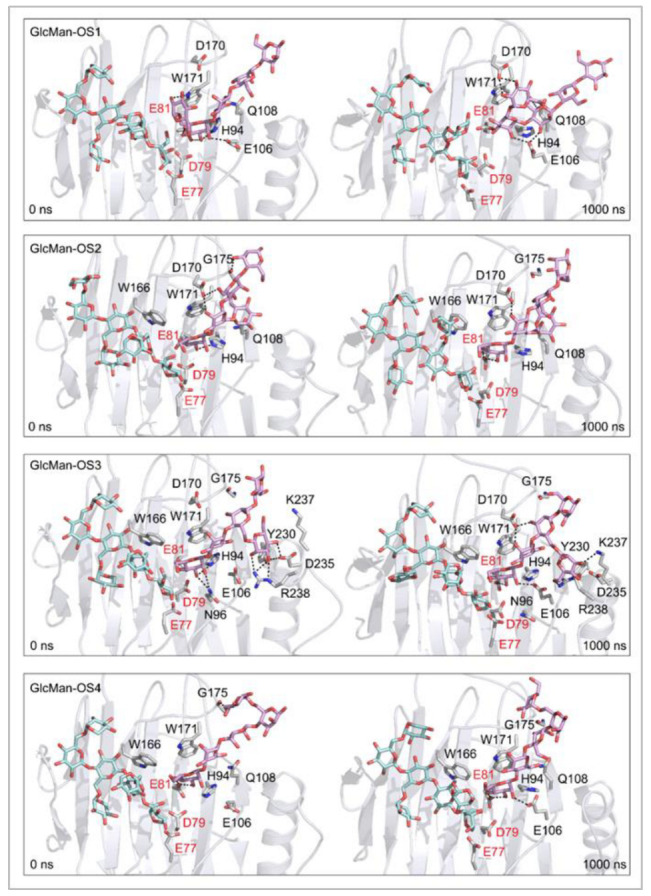
Positions of the XG-OS7 donor and docked GlcMan-OS acceptor substrates that interact with the TmXET6.3 residues (interactions shown at the acceptor sites). The residues that interact with acceptors over 50% of MD simulation times at distances of up to 4.0 Å (right, black letters) are shown at the beginning (0 ns) and after 1000 ns of MD simulation times. Catalytic residues are shown in red letters.

**Table 1 ijms-23-11838-t001:** Abbreviations and structural formulae of the donor and a variety of acceptor oligosaccharide substrates.

Substrate	Name/Abbreviation	Structural Formula
Donor/Acceptor	Xyloglucan heptaoligosaccharide/XG-OS7	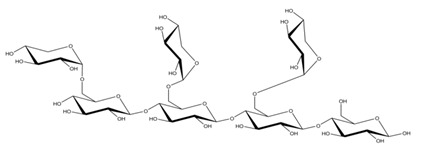
Acceptor	Xyloglucan octaoligosaccharide/XG-OS8	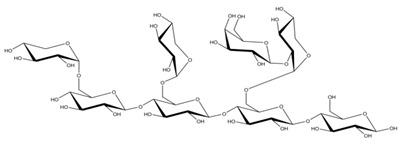
Acceptor	Xyloglucan nonaoligosaccharide/XG-OS9	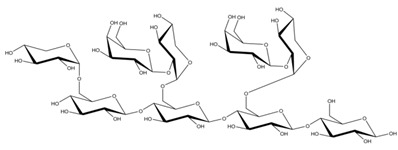
Acceptor	Cellotetraose/Cello-OS4	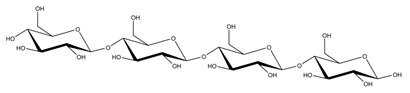
Acceptor	Laminaritetraose/La-OS4	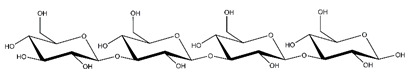
Acceptor	Pustulotetraose/Pu-OS4	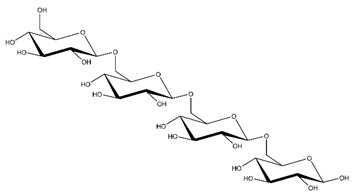
Acceptor	Mannotetraose/Man-OS4	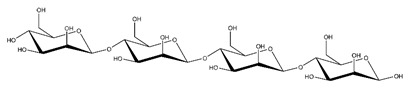
Acceptor	Xylotetraose/Xyl-OS4	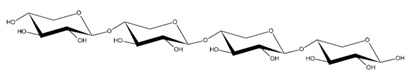
Acceptor	3^3^-β-d-Glucosyl-cellotriose/MLG-OS4A	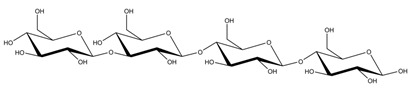
Acceptor	3^2^-β-d-Cellobiosyl-cellobiose/MLG-OS4B	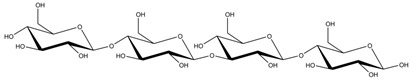
Acceptor	3^1^-β-d-Cellotriosyl-glucose/MLG-OS4C	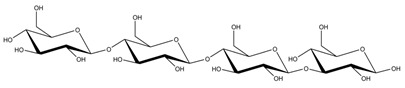
Acceptor	6^4^-α-d-Glucosyl-mannotetraose/ GlcMan-OS1	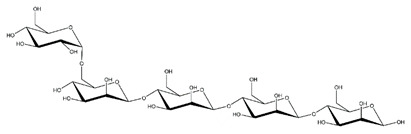
Acceptor	6^3^-α-d-Glucosyl-mannotetraose/ GlcMan-OS2	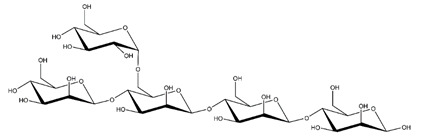
Acceptor	6^2^-α-d-Glucosyl-mannotetraose/GlcMan-OS3	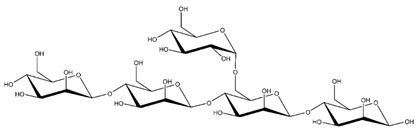
Acceptor	6^1^-α-D-Glucosyl-mannotetraose/GlcMan-OS4	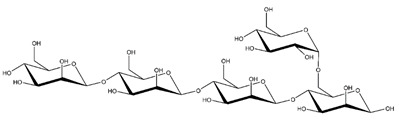
Acceptor	3^3^-α-L-Arabinofuranosyl-xylotetraose/AraXyl-OS	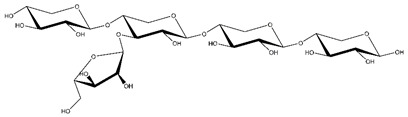

**Table 2 ijms-23-11838-t002:** The TmXET6.3 and PttXET16A residues that interact with XG-OS acceptors over 50% of MD simulation times at distances of up to 4.0 Å.

Acceptor	Interacting Residues (TmXET6.3)	Interacting Residues (PttXET16A)
XG-OS7	D79, E81, H94, N96, E106, Q108, W166, A168, D170, W171, G175, Y230	E89, Q102, E114, R116, W174, A176, D178, W179
XG-OS8	D79, E81, H94, N96, E106, Q108, W166, D170, W171, G175	D87, E89, Q102, N104, E114, R116, W174, A176, W179, G183, R258
XG-OS9	D79, E81, H94, N96, F98, E106, Q108, W166, A168, W171, G175, Y230	E89, N104, F106, E114, R116, W174, W179, G183, R258, Y250

**Table 3 ijms-23-11838-t003:** Binding free energy values of TmXET6.3 and PttXET16A in complex with XG-OS acceptors based on MM(PB/GB)SA calculations. Values were obtained from 800–1000 ns time intervals of MD simulations.

	TmXET6.3				PttXET16-34A			
	MMPBSA		MMGBSA		MMPBSA		MMGBSA	
Acceptor	E_int_ [kcal/mol]	± σ	E_int_ [kcal/mol]	± σ	E_int_ [kcal/mol]	± σ	E_int_ [kcal/mol]	± σ
XG-OS7	−41.36	8.44	−44.68	7.56	−46.12	8.48	−54.91	7.99
XG-OS8	−65.99	9.04	−64.64	8.76	−46.2	9.75	−53.7	9.54
XG-OS9	−70.57	8.57	−69.74	7.66	−74.99	7.52	−76.79	7.07

**Table 4 ijms-23-11838-t004:** The TmXET6.3 and PttXET16A residues that interact with linear acceptors over 50% of MD simulation times at distances of up to 4.0 Å.

Acceptor	Interacting Residues (TmXET6.3)	Interacting Residues (PttXET16A)
Cello-OS4	D79, E81, H94, N96, F98, E106, W166, W171, G175	E89, Q102, N104, E114, R116, W179
La-OS4	D79, E81, H94, N96, E106, Q108, W166, W171, G175, R238	D87, E89, Q102, D104, E114, R116, W174, W179
Man-OS4	D79, E81, H94, N96, E106, W171, G175	I100, R116, T181, R182, G183
Pu-OS4	H94, N96, E106, Q108, W166, D170, W171, G175	Q102
Xyl-OS4	E81, H94, N96, E106, W166, W171, G175	D87, E89, Q102, W174, D178, W179

**Table 5 ijms-23-11838-t005:** Binding free energy values of TmXET6.3 and PttXET16A in complex with linear acceptors based on MM(PB/GB)SA calculations. Values were obtained from 800–1000 ns time intervals of MD simulations.

	TmXET6.3				PttXET16-34A			
	MMPBSA		MMGBSA		MMPBSA		MMGBSA	
Acceptor	E_int_ [kcal/mol]	± σ	E_int_ [kcal/mol]	± σ	E_int_ [kcal/mol]	± σ	E_int_ [kcal/mol]	± σ
Cello-OS4	−48.45	5.35	−49.02	4.74	−26.5	6.27	−28.24	6.23
La-OS4	−43.50	7.49	−45.61	6.25	−41.06	6.84	−41.80	6.31
Man-OS4	−32.15	10.49	−39.32	11.61	−38.26	5.4	−37.19	4.86
Pu-OS4	−45.17	7.85	−45.37	6.87	−32.62	5.9	−33.52	5.93
Xyl-OS4	−31.87	7.31	−33.84	6.55	−23.38	5.53	−24.07	6.15

**Table 6 ijms-23-11838-t006:** The TmXET6.3 residues that interact with MLG-OS acceptors over 50% of MD simulation times at distances of up to 4.0 Å.

Acceptor	Interacting Residues (TmXET6.3)
MLG-OSA	-
MLG-OSB	E81, H94, N96, E106, W166, W171, Y230
MLG-OSC	D79, E81, H94, N96, E106, W166, W171

**Table 7 ijms-23-11838-t007:** Binding free energy values of TmXET6.3 in complex with MLG-OS acceptors based on MM(PB/GB)SA calculations. Values were obtained from 800–1000 ns time intervals of MD simulations.

	MMPBSA		MMGBSA	
Acceptor	E_int_ [kcal/mol]	± σ	E_int_ [kcal/mol]	± σ
MLG-OSA	−33.34	9.15	−32.16	9.72
MLG-OSB	−42.62	7.22	−40.46	6.25
MLG-OSC	−41.64	6.85	−29.50	7.83

**Table 8 ijms-23-11838-t008:** The TmXET6.3 residues that interact with AraXyl-OS and GlcMan-OS acceptors over 50% of MD simulation times at distances of up to 4.0 Å.

Acceptor	Interacting Residues (TmXET6.3)
AraXyl-OS	D79, E81, H94, N96, E106, Q108, W166, W171
GlcMan-OS1	E81, H94, E106, Q108, D170, W171
GlcMan-OS2	D79, E81, H94, Q108, W166, D170, W171, G175
GlcMan-OS3	D79, E81, H94, N96, E106, W166, D170, W171, G175, Y230, D235, K237, R238
GlcMan-OS4	D79, E81, H94, E106, Q108, W166, W171, G175

**Table 9 ijms-23-11838-t009:** Binding free energy values of TmXET6.3 in complex with AraXyl-OS and GlcMan-OS acceptors based on MM(PB/GB)SA calculations. Values were obtained from 800–1000 ns time intervals of MD simulations.

	MMPBSA		MMGBSA	
Acceptor	E_int_ [kcal/mol]	± σ	E_int_ [kcal/mol]	± σ
AraXyl-OS	−45.32	6.44	−43.50	6.53
GlcMan-OS1	−28.45	7.22	−27.51	7.5
GlcMan-OS2	−47.28	6.62	−58.42	6.47
GlcMan-OS3	−32.3	9.53	−35.08	11.23
GlcMan-OS4	−42.75	9.97	−47.32	9.35

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
