# Peer review of "Definition of the Acceptor Substrate Binding Specificity in Plant Xyloglucan Endotransglycosylases Using Computational Chemistry"

_ijms, 2022, doi:10.3390/ijms231911838_

Round 1
Reviewer 1 Report
This manuscript by Stratilová et al. emplyed compuational chemistry approach to define the acceptor substrate biding spectrum in plant xyloglycan endotransglycisylases (XETs). Although the reuslts were demonstrated and explained nicely, this readers might still prefer some experimental results. Without the verification of the binding affinity between the XETs and substrats, the authors should tone down some of the conlusions along with more citable results from other literature. Still, this paper is well-written and suitable for publication with minor revision if the authors can provide some results from real chemical experiments.
Author Response
Dear Reviewer,
please find attached our manuscript entitled ‘Definition of the acceptor substrate binding specificity in plant xyloglucan endotransglycosylases using computational chemistry’, by Barbora Stratilová, Eva Stratilová, Maria Hrmova, and Stanislav Kozmon.
We have revised the manuscript and taken into consideration all your comments.
Specifically, as advised, we have revised the entire text, included a new Reference [63], and extended the Conclusions section. The changes are highlighted in green.
Thank you for your report.
With best wishes,
Maria Hrmova

Reviewer 2 Report
This study investigated the basis of acceptor substrate binding specificity in non-specific Tropaeolum majus (TmXET6.3) and specific Populus tremula x tremuloides (PttXET16A) XETs, using molecular docking and molecular dynamics (MD) simulations combined with binding free energy calculations. The findings in this study explain the differences in acceptor-binding specificity between non-specific and specific XETs and associate theoretical to experimental data. As the authors claim, XET enzymes will contribute not only to XG but also to other biopolymers integrations into plant CWs and potentially to the synthesis of the XG-hybrid molecules. Therefore, I recommend the publication of this manuscript after the following minor revisions.
1. In Table 1, some acceptor structures are not entirely drawn. Please revise.
2. English throughout the text should be checked and revised.
Author Response
Dear Reviewer,
please find attached our manuscript entitled ‘Definition of the acceptor substrate binding specificity in plant xyloglucan endotransglycosylases using computational chemistry’, by Barbora Stratilová, Eva Stratilová, Maria Hrmova, and Stanislav Kozmon.
We have revised the manuscript and taken into consideration all your comments.
Specifically, as advised, we have revised the entire text (checked the English syntax), revised Tables 1 and 3, included a new Reference [63], and extended the Conclusions section. The changes are highlighted in green.
Thank you for your report.
With best wishes,
Maria Hrmova
